# Interconnected Mechanistic Pathways, Molecular Biomarkers, and Therapeutic Approach of Oral Cancer in Patients with Diabetes Mellitus

**DOI:** 10.3390/cimb47110929

**Published:** 2025-11-07

**Authors:** Viviana Elian, Violeta Popovici, Mihnea Ioan Nicolescu, Alexandra Maria Nicolescu, Sorina Maria Aurelian, Emma Adriana Ozon

**Affiliations:** 1Nutrition and Metabolic Disease Unit, “Carol Davila” University of Medicine and Pharmacy, 020475 Bucharest, Romania; viviana.elian@umfcd.ro; 2Diabetes, Nutrition and Metabolic Disease Unit, National Institute of Diabetes, Nutrition and Metabolic Disease, Prof. N. C. Paulescu, 020475 Bucharest, Romania; 3Center for Mountain Economics, “Costin C. Kiritescu” National Institute of Economic Research (INCE-CEMONT), Romanian Academy, 725700 Vatra-Dornei, Romania; 4Division of Histology, Faculty of Dentistry, “Carol Davila” University of Medicine and Pharmacy, 050474 Bucharest, Romania; mihnea.nicolescu@umfcd.ro; 5Laboratory of Radiobiology, “Victor Babeș” National Institute of Pathology, 050096 Bucharest, Romania; 6Immunogenetics and Virology Center, Fundeni Clinical Institute, 022328 Bucharest, Romania; alexandra-maria.balanici@rez.umfcd.ro; 7Clinic of Geriatrics, Hospital of Chronic Diseases “Sf. Luca”, “Carol Davila” University of Medicine and Pharmacy, 020945 Bucharest, Romania; sorina.aurelian@umfcd.ro; 8Department of Pharmaceutical Technology and Biopharmacy, Faculty of Pharmacy, “Carol Davila” University of Medicine and Pharmacy, 020945 Bucharest, Romania; emma.budura@umfcd.ro

**Keywords:** diabetes mellitus, oral cancer, molecular mechanistic pathways, molecular biomarkers, clinical relevance, antidiabetic drugs, therapeutic targets

## Abstract

The complex bidirectional relationship between diabetes mellitus (DM) and oral cancer (OC) denotes that metabolic dysfunction and malignancy intersect at molecular, cellular, and systemic levels. This state-of-the-art review analyzes the most recent literature data on the multiple interconnected pathways linking DM and OC, including hyperinsulinemia/IGF-1 signaling, chronic hyperglycemia-induced cellular damage, persistent inflammation, immune dysfunction, and oral microbiota dysbiosis. These mechanisms create a permissive environment for oral carcinogenesis while simultaneously impairing the body’s natural tumor surveillance systems. Key molecular networks explored include the PI3K/AKT/mTOR pathway, AGE-RAGE interactions, NF-κB signaling, the p53 tumor suppressor pathway, and HIF-mediated responses. Clinical evidence demonstrates that patients with diabetes have higher OC prevalence (250 per 100,000 patients) and significantly increased mortality (HR of 2.09) compared to non-diabetics. The review highlights metformin as the most promising anti-diabetic agent for OC management, showing anti-tumor effects through mTOR inhibition. Novel therapeutics, such as GLP-1 agonists, particularly semaglutide, may be helpful but require further clinical validation. Understanding the shared molecular pathways enables the development of integrated therapeutic strategies that target both conditions simultaneously, and it supports effective screening programs, personalized prevention strategies, and optimized multidisciplinary management approaches for this high-risk patient population.

## 1. Introduction

Oral cancer ranks as the sixth most prevalent cancer globally, affecting approximately 650,000 individuals annually [1]. The incidence of oral cancer has escalated by 49% over the past decade, according to the Mouth Cancer Foundation (available at https://www.mouthcancerfoundation.org/mouth-cancer-facts-and-figures, accessed on 1 November 2025). Still, two-thirds of patients are diagnosed at an advanced stage of disease, with a 5-year survival rate of 50% or less [2]. Notably, ten-year survival rates exhibit considerable variability, ranging from 18% to 57%, and are contingent on the cancer’s specific location and the timeliness of diagnosis [3]. It presents substantial mortality and morbidity challenges, particularly in Southeast Asia, South Asia, and certain regions of Europe, where incidence rates are highest [4,5,6,7]. The primary risk factors associated with oral cancer include tobacco and alcohol consumption; however, emerging research suggests that obesity may also be a significant contributing factor to the increased risk of developing this disease [8].

Obesity is also a significant risk factor for developing diabetes mellitus (DM), as it leads to insulin resistance and impairs the function of insulin-producing cells; when excess body fat accumulates, particularly around the abdomen, it raises levels of fatty acids and other substances that decrease the body’s sensitivity to insulin [9]. Affecting over 400 million people worldwide, DM has emerged as a potential risk factor for various cancers [10]. The relationship between diabetes and cancer has been recognized for over a century. Still, recent epidemiological and molecular studies have provided more concrete evidence for this association, particularly regarding oral malignancies [11,12,13].

The presence of DM encompasses increased cancer risk, altered treatment outcomes, and heightened complications, making OC a particular burden for diabetic patients [13,14]. Patients with DM have a higher risk of OC development than non-diabetic patients (Odds Ratio = 1.41); a pooled prevalence of 0.25% (250 per 100,000 patients) in diabetic patients was reported [15,16]. The association is influential for precancerous lesions; leukoplakia reveals a pooled prevalence of 2.49% among diabetic patients [15]. A Danish population-based cohort study found that the incidence of OC was increased 2-fold in the diabetic population younger than 50 years [17]. Austrian data revealed that 59.9% of OC patients had glucose metabolism disorder compared to 36.5% in controls [18]. In Austria, numerous people have diabetes (7–11% of the total population), with OC being the tenth most common malignancy [18]. In Hungary, oral cavity cancer shows the second-highest age-standardized mortality rate across European nations, with approximately 3000 new diagnoses and more than 1600 deaths annually [19].

Patients with OC and DM had significantly higher mortality than controls, with an HR of 2.09 (95%CI = 1.36–3.22) [15]. Diabetic patients with oral squamous cell carcinoma (OSCC) tend to have lower overall survival, recurrence-free survival, and cancer-specific survival compared with non-diabetics [20]. The negative impact persists even in less aggressive tumor stages (stage I and II), with diabetic patients who were not prescribed adjuvant therapy showing a significantly higher recurrence rate than non-diabetic patients [20]. DM significantly impacts the OC development, metastasis, and prognosis. This influence may be attributed to several associated factors (hyperglycemia, hyperinsulinemia, insulin resistance, and chronic inflammation) [21]. The bidirectional relationship is further complicated by periodontal disease negatively impacting glycemic control and potentially contributing to diabetes complications [19]. Effective management of DM in patients undergoing OC treatment requires a multidisciplinary approach. It should include a diabetes specialist, educator, dietician, pharmacist, and psychosocial support professional, who collaborate closely with the oncology care team. Such comprehensive management is essential for preventing diabetes-related complications and ultimately improving patient outcomes [22,23].

Significant clinical and epidemiological studies, along with their corresponding findings, are summarized in Table 1.

Considering the potentially harmful effects of oral cancer, all these findings are worrying. The evidence clearly establishes OC as a significant burden for diabetic patients, characterized by increased incidence, worse prognosis, and heightened treatment complications [15]. Ongoing research is examining pathogenic pathways that explain the association between diabetes and OC [17]. As long as it remains unknown why patients with diabetes are at higher risk for oral cancer, targeted prevention and treatment programs are unlikely to succeed [17]. Therefore, this comprehensive analysis examines the most recent findings regarding key molecular pathways involved in both diseases, their dysregulation patterns, cross-talk mechanisms, and therapeutic implications. It examines the progress of current and emerging molecular targets, clinical significations, therapeutic approaches, and clinical developments that could enrich management strategies for patients with these interconnected conditions.

## 2. Biological Mechanisms That Correlate Diabetes Mellitus with Oral Cancer

Beyond incidence, DM also significantly impacts the prognosis of patients with oral cancer. Studies have consistently shown that: (a) patients with OC and DM have higher mortality compared to non-diabetic patients with oral cancers; (b) diabetic patients tend to have lower overall survival, recurrence-free survival, and cancer-specific survival; (c) even in less aggressive tumor stages (stages I and II), diabetic patients demonstrate worse outcomes [17,20,21].

The association between DM and OC involves multiple, interconnected biological mechanisms (Figure 1).

### 2.1. Hyperglycemia-Mediated Effects

High glucose concentrations directly impact cancer cells through: (a) metabolic reprogramming (providing enhanced energy substrate for rapidly dividing cancer cells) [28]; (b) oxidative stress (hyperglycemia increases reactive oxygen species (ROS) production through mitochondrial metabolism, leading to DNA damage and mutations); (c) signaling pathway activation (high glucose activates various pathways controlling proliferation, migration, invasion, and recurrence); (d) hyperglycemic memory (cancer cells exposed to hyperglycemic conditions maintain permanently activated oncogenic pathways even after glucose normalization, suggesting epigenetic modifications that persist beyond the initial metabolic insult) [29,30,31,32,33].

### 2.2. Hyperinsulinemia

Insulin resistance (IR) results in compensatory hyperinsulinemia. This elevated insulin level may directly promote the proliferation of OC cells. High insulin levels can also promote tumor cell growth by stimulating DNA synthesis and inhibiting apoptosis, thereby facilitating cancer cell invasion and metastasis [34,35,36].

### 2.3. Chronic Inflammation

DM has increasingly emerged as a condition distinguished by chronic low-grade systemic inflammation. This persistent inflammatory state is predominantly driven by metabolic stresses arising from hyperglycemia and insulin resistance. Chronic inflammation plays a pivotal role in facilitating carcinogenesis through various intricate mechanisms. Inflammatory cytokines can instigate oxidative stress and DNA damage, significantly elevating the risk of mutagenesis and genomic instability. Moreover, these mediators not only promote cellular proliferation but also inhibit apoptosis and stimulate angiogenesis, thereby creating a microenvironment highly conducive to tumor development and growth. Within oral tissues, inflammatory signaling can alter the behavior of keratinocytes and immune cells, thereby enhancing the potential for malignant transformation, particularly in combination with other carcinogenic factors such as tobacco or alcohol [12,15,37,38,39,40].

### 2.4. Immune Dysfunction

Patients with DM often encounter a range of health challenges due to compromised immune surveillance mechanisms, altered cytokine profiles, and a reduced capacity for OC cells recognition and elimination [13,41].

### 2.5. Angiogenesis Dysregulation

The pro-angiogenic state associated with DM may significantly influence the onset and progression of various complications related to diabetes. This altered physiological condition not only impacts vascular integrity but also contributes to the initiation and aggressive behavior of oral neoplastic lesions. Enhanced angiogenesis can foster a favorable microenvironment for OC development, potentially leading to more severe health outcomes in individuals with DM [42,43].

### 2.6. Oral Microbiota Dysbiosis

Recent research has revealed a complex, bidirectional relationship between oral pathogens and systemic diseases, particularly DM and OC [44,45,46]. For example, periodontal disease can significantly disrupt glycemic control, making it more difficult for individuals to manage their blood sugar levels effectively. This deterioration in oral health not only affects the mouth but also contributes to systemic issues, as hyperglycemia creates an environment conducive to the proliferation of harmful oral pathogens. These pathogens thrive in high-glucose conditions, perpetuating a destructive cycle where poor oral health exacerbates DM outcomes, leading to further complications. In addition to this reciprocal relationship, specific strains of oral bacteria have been identified as producers of carcinogenic metabolites. These compounds, along with the chronic inflammation triggered by oral pathogens, can initiate cellular changes that lead to malignant transformation [47].

## 3. Molecular Pathways in Diabetes Mellitus and Oral Cancer

All the abovementioned data indicate that DM and OC share multiple biological and molecular pathways, forming a complex network of pathophysiological interactions. Having previously established the epidemiological association and biological mechanisms linking DM to OC (hyperglycemia- and hyperinsulinemia-mediated effects, chronic inflammation, immune dysfunction, angiogenesis dysregulation, and oral microbiota dysbiosis), this section provides a detailed molecular description of the key pathways involved, including their normal physiological functions, alterations in diabetes and oral cancer, and points of convergence between both diseases. Understanding these molecular details is essential for identifying therapeutic targets and developing biomarker-based risk stratification strategies.

### 3.1. AGE-RAGE Interactions: Molecular Mechanisms and Implications in Diabetes and Oral Cancer

In diabetes, hyperglycemia-induced signaling pathways lead to insulin resistance and β-cell dysfunction [48]. High blood glucose levels accelerate the formation and accumulation of advanced glycation end products (AGEs) in tissues and the bloodstream. They result from a glucose non-enzymatic reaction (glycation) with proteins, lipids, or nucleic acids [49]. The AGE-RAGE signaling pathway is a complex molecular cascade triggered when AGEs bind their cognate receptors (RAGEs) [50]. This binding initiates various downstream signaling pathways, leading to cellular responses that contribute to diseases such as diabetes and its complications, including OC [51]. RAGEs are overexpressed in OC associated with increased invasive and metastatic activity. Key downstream effects include the activation of inflammatory and profibrotic pathways, such as NF-κB, MAPK, and JNK, increased oxidative stress, and alterations in PI3K/AKT signaling [52,53,54]. Through angiogenesis and transcription factor (TF) activation, the AGE-RAGE interaction initiates DM complications, OC progression, invasion, and metastases [55]. Thus, the AGE-RAGE axis serves as a critical molecular bridge between diabetes and oral cancer, facilitating inflammatory and oncogenic processes through complex signaling networks [56] (Table 2).

### 3.2. Insulin/IGF-1 System in Diabetes Mellitus and Oral Cancer

Insulin-like growth factors (IGF-1 and IGF-2) are polypeptide hormones that share a high degree of structural similarity with insulin, the key regulator of glucose metabolism. IGF-1, in particular, is notable for its strong affinity for its specific receptors, known as IGF-1 receptors (IGF-1R) [57]. The primary site of IGF-1 production is the liver, where hepatocytes synthesize and secrete it into the bloodstream. The production of IGF-1 is predominantly regulated by growth hormone (GH), which stimulates its synthesis in response to physiological needs such as growth and tissue repair [44].

Under physiological conditions, IGF-1 binds to both IGF-1R and the insulin receptor (IR), forming the IGF-1/insulin receptor complex. This interaction facilitates complex signaling dynamics that maintain metabolic homeostasis, highlighting the critical interplay between the insulin/IGF-1/GH signaling pathways in ensuring proper growth and metabolic function across diverse tissues [57,58].

IGFs are essential regulators of cell growth and exhibit complex relationships with oral cancer. Several studies suggest increased expression of IGF-2 and the IGF-1 receptor in oral tumors, suggesting a potential role in OC proliferation; other research has demonstrated no significant difference in IGF-1 expression levels between healthy tissues and those affected by OC [59,60,61]. Upon binding to IGF-1R, IGF-1 initiates the activation of two critical intracellular signaling pathways: the p38 mitogen-activated protein kinase (MAPK), protein kinase B (AKT), and the phosphoinositide 3-kinase (PI3K) pathways (Table 2). These pathways are essential for various tumor cellular processes (proliferation, differentiation, and survival) [55,62,63]. Insulin/IGF-1 signaling contributes to obesity-associated cancer risk [64,65,66]. Understanding these mechanisms can potentially lead to better OC management (Table 3).

### 3.3. Core Molecular Pathways

The most essential molecular pathways shared between DM and OC are the PI3K/AKT/mTOR pathway, the inflammatory pathway, the p53 tumor suppressor network, the HIF-mediated pathway, and the oral microbiome pathway.

#### 3.3.1. PI3K/AKT/mTOR Signaling Network

The PI3K/AKT/mTOR pathway represents a central hub for metabolic regulation and cellular growth control, making it essential in both DM and OC pathogenesis [68,69] (Table 4). The phosphoinositol 3-kinase (P13K) and mechanistic target of rapamycin (mTOR) play essential roles in cell homeostasis and metabolism [70,71]. The PI3K/Akt/mTOR signaling pathway is dysregulated in DM [70,72]. It mediates OC progression and chemotherapy resistance [73,74].

PTEN is a key tumor suppressor that helps control cell growth, survival, and division. It does this by dephosphorylating PIP3 (phosphatidylinositol (3,4,5)-trisphosphate), thereby counteracting PI3K/Akt signaling. PTEN plays essential roles in cell movement, adhesion, and maintaining genomic stability [75]. Therefore, PTEN is vital for maintaining the structural and functional integrity of pancreatic islets, which produce insulin and regulate insulin signaling and glucose metabolism. When PTEN is dysfunctional, it can lead to type 2 DM. Changes in PTEN activity can hinder insulin’s ability to manage glucose uptake and balance in tissues such as the liver, muscle, and pancreas [76]. On the other hand, mutations that reduce PTEN activity may increase insulin sensitivity, but they also raise the risk of obesity and cancer [77]. PTEN loss contributes to OC development and progression [78] (Table 4)

**Table 4 cimb-47-00929-t004:** Architecture and functions of the molecular pathways shared by DM and OC.

Molecular Players	Normal Function	DM Implications	OC Implications	References
**PI3K/AKT/mTOR signaling network**				
PI3K	- Cell survival - Cell metabolism - Glucose uptake - Immune system function	- Insulin resistance - Beta-cell dysfunction	- Oncogenic activation - Tumor progression	[79,80]
Protein kinase B (AKT)	- Metabolic regulation	- Impaired glucose transport	- Enhances tumor cell survival	[79,81]
mTOR	- Metabolic regulation, - Cell homeostasis	- Insulin resistance - DM complications	- Tumor proliferation - Metastases	[82]
PTEN	- Tumor suppressor - Regulator of glucose metabolism	- Reduced insulin activity	- Tumorigenesis	[83]
**Inflammatory pathway**				
NLRP3	- Sensor in the innate immune system - Insulin release regulation - Beta-cell proliferation	- Hyperglycemia sensing	- Tumor-promoting inflammation	[84,85]
ASC	- Adapter protein for inflammasome formation	- Inflammasome assembly - Insulin resistance - Pancreatic cell dysfunction	- Cancer cell proliferation - OC invasion	[86]
Caspase-1	- Protease activation	- IL-1β/IL-18 processing	- Tumor microenvironment	[87]
IL-1β	- Pro-inflammatory cytokine	- Insulin resistance	- Angiogenesis promotion	[87]
IL-18	- Immune activation	- β-cell dysfunction	- Immune evasion	[88,89]
**p53 tumor suppressor network**				
p53	- Transcription factor - Glucose metabolism	- Metabolic stress - Insulin resistance	- Tumor progression	[90]
MDM2	- Negative regulator of p53	- Insulin resistance - DM complications	- Oncogenic activation - OC development	[91]
p21	- Cell cycle inhibitor	- Insulin resistance	- Oncogenic role	[92,93]
BAX/BAK	- Inactive in the absence of a stress signal	- β-cell death	- Cancer cell death	[94,95]
**Hypoxia-Inducible Factor (HIF) pathway**				
HIF-1α	- Hypoxia response	- Diabetic complications	- Tumor angiogenesis	[96,97]
HIF-2α	- Chronic hypoxia	- Vascular dysfunction	- Metastasis	[98,99]
VEGF	- Angiogenesis	- Diabetic retinopathy	- Tumor vascularization	[43,100]
GLUT1	- Glucose transport	- Metabolic adaptation	- Warburg effect	[101]
**Oral microbiome pathway**				
***Microbial* sp.**	**Pathway targets**	**Diabetes implications**	**OC implications**	**References**
*Fusobacterium nucleatum*	TLR4/NF-κB, Wnt/β-catenin	Inflammation	Tumor promotion	[102,103]
*Porphyromonas gingivalis*	NLRP3, PI3K/AKT	Insulin resistance	DNA damage, tumor induction, and progression	[103,104]
*Candida albicans*	NF-κB, complement	Immune dysfunction	DNA damage	[103,105]

#### 3.3.2. Inflammatory Pathway

Inflammatory mediators create a permissive environment for both metabolic dysfunction and tumor development (Table 4). NOD-like receptor (NLR) family pyrin domain-containing 3 (NLRP3) has a complex role in the innate immune system and cell metabolism [106]. The NLRP3 inflammasome serves as a molecular platform that detects metabolic stress and pathogen-associated signals, triggering inflammatory responses that lead to DM complications and the progression of OC [107].

In diabetes, persistent hyperglycemia and metabolic dysregulation lead to chronic NLRP3 activation in pancreatic β-cells, adipose tissue, and vasculature [107]. Moreover, metabolic stressors such as hyperglycemia, hyperlipidemia, and free fatty acids can trigger inflammasome activation [39]. Free fatty acids and advanced glycation end products, mitochondrial dysfunction, and oxidative stress also activate these mechanisms (Table 4).

In OSCC, NLRP3 contributes to cancer cell proliferation, epithelial–mesenchymal transition (EMT), and enhanced metastatic potential by activating NF-κB and STAT3 signaling pathways [108]. Carcinogen exposure, DNA damage, tumor microenvironment factors, oral microbiota dysbiosis, and bacterial components activate NLRP3 inflammasome signaling [109].

Apoptosis-associated speck-like protein containing a CARD (ASC) is a central adaptor protein for inflammasome formation [110]. The ASC inflammasome stands as a pivotal player in the onset of diabetes, intricately linking chronic inflammation to the troubling phenomena of insulin resistance and pancreatic dysfunction. Acting as a vital adapter protein, it skillfully assembles with notable partners such as NLRP3 and procaspase-1 to detect metabolic stress signals—including free fatty acids, cholesterol, and islet amyloid polypeptide —substances all too common in the landscape of diabetes. This assembly triggers the activation of pro-inflammatory cytokines, particularly IL-1β and IL-18, which not only fuel insulin resistance but also damage pancreatic beta cells and intensify the complications associated with DM [111,112]. Studies using OSCC cell lines have shown that high ASC expression levels enhance cell migration and invasion. In vivo experiments have confirmed that ASC overexpression promotes the metastasis of OSCC cells [113]. Caspases and inflammatory cytokines IL1β and IL18 are overexpressed in DM and OC development [114,115,116,117] (Table 4)

#### 3.3.3. p53 Tumor Suppressor Network

The p53 tumor suppressor pathway integrates multiple stress signals and coordinates cellular responses, including DNA repair, cell cycle arrest, and apoptosis [118]. The p53 mutation (TP53) is responsible for the onset and progression of both DM and OC [119,120,121,122]. Metformin selectively induces apoptosis in cells lacking functional p53 by activating the AMPK pathway, thereby coordinating metabolic stress response and the DNA damage response to hyperglycemia-induced oxidative stress [90]. This pathway also influences cell cycle checkpoint control, DNA repair coordination, and the induction of apoptosis in damaged cells (Table 4). Moreover, dysregulation of the MDM2-p53 pathway affects mitochondrial metabolism and contributes to inflammation and cellular damage in DM and OC [123,124]. The p21 protein is a cell cycle inhibitor [125]. p21 significantly influences the development of diabetes mellitus through multiple mechanisms. It plays an essential role in the function of pancreatic β-cells, which produce insulin to regulate blood sugar levels. In individuals with type 2 DM, p21 levels are often elevated in the pancreas, contributing to inflammation and reducing insulin production [126]. In OC patients, high p21 levels are associated with metastases [127,128]. In a healthy cell, the Bcl-2 proteins BAX and BAK are inactive and spread throughout the cytosol. When stress signals are received, they are transduced and transported to the outer mitochondrial membrane, where they form pores that release pro-apoptotic factors, such as cytochrome c. This release initiates a cascade of events leading to programmed cell death (apoptosis) [129]. In DM, both proteins induce pancreatic cell death [130], while in OC, they induce tumor cell apoptosis [131] (Table 4).

#### 3.3.4. Hypoxia-Inducible Factor (HIF) Pathways

HIF pathways coordinate cellular responses to hypoxia and metabolic stress, playing substantial roles in both diabetic complications and cancer progression [99,132] (Table 4). HIF-1α drives metabolic reprogramming in both diabetes and cancer, promoting aerobic glycolysis. In diabetes, elevated glucose levels induce HIF-1α activity, leading to a switch from oxidative to glycolytic metabolism and a Warburg-like metabolic phenotype. Vascular Endothelial Growth Factor (VEGF) is significantly higher in oral cancer. It plays a key role in promoting tumor growth, proliferation, and metastasis by stimulating the formation of new blood vessels (angiogenesis). Elevated levels of VEGF in DM are linked with DM retinopathy [133], while in OSCC, they have been associated with lymph node metastasis and a poorer prognosis [134]. Research indicates that changes in glucose metabolism occur early in the development of oral cancer. The notable increase in GLUT1 expression [135] is closely linked to lymph node status and is associated with poor outcomes and reduced survival in patients with OC.

#### 3.3.5. Oral Microbiome Pathway

The oral microbiome comprises over 700 bacterial species, with periodontal pathogens playing crucial roles in both local and systemic pathology [136]. Emerging evidence reveals bidirectional relationships between oral pathogens and systemic diseases, particularly DM and OC (Figure 1) [102,137,138]. Therefore, periodontal health is a modifiable risk factor for both conditions [139]. Periodontal disease affects glycemic control while hyperglycemia promotes pathogen proliferation, creating a vicious cycle [140]. Pathogenic oral bacteria produce carcinogenic metabolites and promote chronic inflammation that drives malignant transformation; the neoplastic tissues are rich in most of them [141].

The essential bacterial and fungal pathogens associated with the induction and progression of OC in patients with diabetes are illustrated in Table 4.

##### Gram-Negative Bacteria

Gram-negative periodontal pathogenic bacteria, *P. gingivalis* and *F. nucleatum*, secrete lipopolysaccharides (LPS), which subsequently induce the release of pro-inflammatory cytokines. Among these cytokines are tumor necrosis factor-alpha (TNF-α), interleukin-6 (IL-6), and C-reactive protein (CRP) [103]. These inflammatory mediators have been shown to impair insulin signaling, increase insulin resistance, and exacerbate hyperglycemia. Furthermore, persistent periodontal infections have been strongly correlated with elevated hemoglobin A1c (HbA1c), a widely recognized key indicator of long-term glycemic control [142]. These Gram-negative periodontal pathogens also play a pivotal role in promoting carcinogenesis [143].

*P. gingivalis* disrupts the body’s insulin signaling pathways, leading to insulin resistance and potentially increasing the risk of DM [144,145]. It potentiates stem-like properties of oral squamous cell carcinoma by modulating SCD1-dependent lipid synthesis via the NOD1/KLF5 axis [146]. *P. gingivalis* can promote OC development by inducing epithelial–mesenchymal transition (EMT), altering the tumor immune microenvironment, protecting cancer cells from macrophage attack, and developing drug resistance [147,148,149,150]. It has shown high development in patients with diabetes and OC [151].

*F. nucleatum* accelerates cancer development via the adhesin FadA, triggering the Wnt/β-catenin signaling pathway. Bacterial lipopolysaccharides stimulate TLR4, inducing inflammation and activating NF-κB, while microbial metabolites modulate host pathway activity (Table 3) [152]. *F. nucleatum* possesses the remarkable ability to suppress the activity of natural killer (NK) cells while simultaneously drawing myeloid-derived suppressor cells (MDSCs) to the site of infection. This intricate interplay can indirectly foster the emergence of cancer. The pivotal factor in this process is the bacterial virulence factor FAP2, which adeptly binds to and obstructs the NK receptor TIGIT. As a result, the immune system’s powerful NK cells are unable to mount an effective attack on tumor cells, allowing cancer to gain a foothold [153].

##### *C. albicans* 

DM patients exhibit increased susceptibility to *C. albicans* oral infections, attributed to higher salivary glucose levels, reduced salivary flow, microvascular degeneration, generalized immunosuppression, and impaired neutrophil candidacidal activity [154,155]. Recent studies reported a high occurrence of this opportunistic fungus in periodontal disease; its hyphae were found in the connective tissue of periodontal patients in association with highly invasive anaerobic bacteria, such as *P. gingivalis* [156]. *C. albicans* can directly lead to OC development through carcinogenic enzymes and other compounds (nitrosamines, acetaldehyde) that can cause DNA damage and inhibit DNA repair mechanisms, thus inducing genetic mutations and chromosomal abnormalities [157]. By indirectly binding glutathione, acetaldehyde increases the ROS levels, promotes chronic inflammation, and causes mitochondrial damage [158]. *C. albicans* infection promotes the expression of interleukin-17A (IL-17A) and its receptor (IL-17RA) in OC cells and macrophages. Therefore, the increased IL-17A/IL-17RA signaling activates macrophages, promoting the release of inflammatory cytokines and thereby enhancing the proliferation of OC cells. [159]. *C. albicans* upregulates the expression of programmed death-ligand 1 (PD-L1) in OC cells, leading to an inhibition of T cell activation and proliferation [160]. Moreover, recent studies reported that, by inducing lipid droplet formation, *C. albicans* biofilm may contribute to the development and progression of OC and decrease the efficacy of chemotherapeutic drugs [161].

#### 3.3.6. NF-κB Transcriptional Network

NF-κB serves as a central transcriptional hub coordinating inflammatory responses, metabolic regulation, and cancer progression. Hyperglycemia, insulin resistance, and AGE/RAGE signaling activate the IKK complex, thereby amplifying the production of pro-inflammatory cytokines [54,56]. On the other hand, *F. nucleatum* promotes cancer progression by stimulating TLR4/MyD88 signaling to activate NF-κB, oncogenic pathways, and resistance to apoptosis [162].

Fan et al. identified eight significant genes that were immunologically related cross-talk genes related to OC and T2DM: C1QC (Complement C1q subcomponent, C chain), NOS2 (Nitric Oxide Synthase 2), ABCD1 (ATP-Binding Cassette Sub-Family D Member 1), PD1A4 (PD-1 (Programmed Cell Death Protein 1) Auxiliary Protein 4), ALOX15 (Arachidonate Lipoxygenase 15), CSE1L (Chromosome Segregation 1-Like), IL1RN (Interleukin 1 Receptor Antagonist), and PSMC4 (Proteasome 26S Subunit ATPase 4) [163].

A noteworthy study explored the impact of the LDL (low-density lipoprotein) receptor-related protein 1B polymorphism in diabetic patients diagnosed with OC. It revealed a significant association between the minor allele of rs10496915 and larger tumor size among these patients. Additionally, the heterozygous genotype and specific combinations of minor alleles of rs6742944 were associated with lymph node metastasis and more advanced clinical stages of this neoplastic disease [164]. These findings shed light on the complex ways in which genetic variations influence oral cancer characteristics in individuals with diabetes, thereby enhancing our understanding of this intricate relationship.

A recent study identified four genes (CYP2C19, NLRP3, PVT1, and APP) that appear central to DM’s influence on HNSCC via the protein–protein interaction (PPI) network [165].

### 3.4. Molecular Biomarkers with Clinical Significance in Diabetes and Oral Cancer

Due to their key functions, numerous molecular biomarkers involved in various interconnected pathways of diabetes and OC can be helpful in clinical practice, including diagnosis, prognosis, and monitoring of therapy [117,166,167,168] (Table 5).

## 4. Oral Cancer Treatment in Diabetes Patients

Oral squamous cell carcinoma (OSCC) patients with higher preoperative HbA1c levels had more extended hospitalization and worse survival outcomes [169,170]. While high fasting insulin levels are a significant risk factor for OSCC and are associated with an increased risk of developing the disease, they could be used for the direct early detection of OC [13,172,201]. However, the focus remains on lifestyle changes and monitoring for cancer development in those with signs of insulin resistance [172]. Elevated levels of the protein IGF2BP2 are associated with increased cell proliferation, enhanced cell invasion and metastasis, and altered tumor-infiltrating immune cells in OSCC [202].

Given these metabolic considerations, anti-diabetic drugs are the most promising medications for the treatment of patients with DM and OC [203].

### 4.1. Metformin (Biguanides)

A meta-analysis of 22 randomized controlled trials comprising 5943 participants with various cancer types [204]. It found that the pooled hazard ratios were not statistically significant for progression-free survival (HR, 0.97; 95% CI, 0.82–1.15) and overall survival (HR, 0.98; 95% CI, 0.86–1.13) in patients with cancer between the metformin and control groups [204]. However, metformin appears to be the most promising among the anti-diabetic agents for treating oral cancer. There is evidence from both in vitro and in vivo studies for anti-tumor effects, especially in OSCC.

In vitro and xenograft model experiments revealed that metformin inhibited the growth and metastasis of OSCC cells, with metformin-restrained tumorigenesis accompanied by a substantial decrease in both Aurora-A and Late SV40 Factor (LSF) expressions [205]. It reduced OSCC progression by regulating RNA alternative splicing, promoting NUBP2 splicing to increase the canonical full-length isoform, thereby inhibiting cancer cell proliferation [206]. Moreover, metformin enhanced cisplatin cytotoxicity and reversed chemoresistance in OSCC by inhibiting the NF-κB/HIF-1α signal axis and downregulating hypoxia-regulated genes, potentially serving as a chemosensitizer for cisplatin-based regimens [207].

Clinical trial results suggest that patients with diabetes treated with metformin had a lower incidence of head and neck cancer, improved overall survival, and cancer-specific survival compared to both non-diabetic patients and patients not treated with metformin [208]. There remains high heterogeneity among the study results.

An open-label Phase IIa clinical trial in 35 individuals with oral premalignant lesions (mild to moderate dysplasia) demonstrated that 60% of participants exhibited histological responses after 3 months of metformin treatment, including 17% with complete responses. It occurred concomitant with reduced cell proliferation and modulation of the mTOR pathway, without affecting circulating glucose or C-peptide levels, and independently of participants’ BMI. A significant correlation was found between decreased mTOR activity and both histological (*p* = 0.04) and clinical (*p* = 0.01) responses [209]. This mechanism was further explored, and results suggested that metformin prevented the development of OSCC by significantly reducing the size and number of carcinogen (4-nitroquinoline-1-oxide, 4NQO)-induced oral premalignant lesions and preventing their spontaneous conversion into OSCC in mice in vivo [210].

A study by Madera et al. demonstrated that high OCT3 expression in oral premalignant and malignant lesions enables metformin to inhibit mTOR signaling, with specific effects on mTORC1 and OSCC progression, both in vitro and in vivo. It provides a mechanistic basis for patient stratification in metformin chemoprevention trials [211].

A cohort study with propensity score analysis included 138 patients with T2DM and OSCC who underwent radical surgical treatment and investigated the impact of T2DM and metformin on recurrence and survival outcomes. Results showed that T2DM was associated with a higher risk of OSCC recurrence; however, metformin treatment significantly reduced this increased recurrence risk compared to diabetic patients not receiving metformin, both before (*p* = 0.005) and after (*p* = 0.002) propensity score matching. Apparently, metformin treatment can counteract the negative prognostic impact of T2DM on OSCC recurrence [212].

Metformin treatment reduced the proliferative capacity of OSCC cells by decreasing mTOR and AKT activity, activating AMPK, and suppressing cancer stemness gene-expression programs, while enhancing their commitment to terminal differentiation. The findings revealed that metformin’s anti-tumor effects are concentration-dependent and occur at physiologically relevant doses, providing mechanistic evidence that metformin directly targets carcinoma-initiating cells and supporting its potential for stratified patient selection in OSCC prevention and treatment trials [213].

Based on all available evidence, metformin could serve as a potential adjuvant therapy in HNSCC [214].

### 4.2. GLP-1 Agonists

Obesity is increasingly recognized as a significant cause of cancer in the United States and around the world. However, no medications have been proven to reduce the cancer risk associated with obesity. Recent studies aimed to address this gap by evaluating GLP-1 receptor agonists, widely prescribed to treat diabetes, obesity, and related conditions. The findings suggest that GLP1-RAs may modestly reduce the risk of obesity-related cancers and lower overall mortality [215,216]. While the reported results are encouraging, further studies are needed to establish a clear cause-and-effect relationship.

#### 4.2.1. Semaglutide

A recent study by Wang et al. explored the potential use of semaglutide as a treatment for oral squamous cell carcinoma and the mechanisms underlying its benefits [217]. The study utilized 10 samples of OSCC tissue, 10 samples of normal oral epithelial tissue, and CFPAC-1 cells (human pancreatic cancer cells). In cell culture experiments, semaglutide demonstrated dose-dependent anticancer activity, reducing proliferation markers (Ki-67 and PCNA) and inhibiting epithelial–mesenchymal transition, as well as OSCC cell proliferation [217]. Additionally, semaglutide increased E-cadherin expression and decreased Vimentin expression, thereby suppressing migration, invasion, and epithelial–mesenchymal transition in OSCC cells. In xenograft mouse models, semaglutide significantly reduced tumor volume and weight by increasing E-cadherin, Bax, and GLP-1R expression and decreasing Ki-67, PCNA, Vimentin, and Bcl-xL [217]. It confirms the in vitro findings. Also, semaglutide induced specific activation of the P38 MAPK pathway, leading to increased apoptosis in OSCC tissue [217]. The authors note that clinical validation is essential to confirm therapeutic potential and long-term safety, and future work should explore the combination of semaglutide with existing OSCC treatments, such as chemotherapy or immunotherapy [217].

#### 4.2.2. Liraglutide

Liraglutide has shown anti-tumor effects in several cancer types (hepatic, prostate, pancreatic, breast, colorectal, thyroid, lung, and endometrial) [218,219,220,221,222,223,224,225,226,227,228,229,230]. However, Liu et al. found that liraglutide promoted breast cancer through NOX4/ROS/VEGF pathway activation, directly contradicting other breast cancer studies [224]. There is no published research on its use specifically in OCs. The concentration-dependent effects and contradictory findings in different cancer types suggest caution is needed, and clinical trials would be necessary before considering it for OC treatment.

While semaglutide shows promising preclinical activity against OSCC via p38 MAPK-mediated apoptosis, and liraglutide demonstrates anticancer effects in multiple malignancies (with notable contradictions), the complete absence of clinical data specific to OC precludes any therapeutic recommendations supporting GLP-1 agonist use for oral cancer.

#### 4.2.3. SGLT Inhibitors

SGLT2 inhibitors have emerged as promising agents in the battle against cancer, demonstrating notable anticancer effects across various malignancies, including breast, liver, pancreatic, thyroid, prostate, and lung cancers. Their efficacy is attributed to several mechanisms: they induce mitochondrial membrane instability, inhibit key signaling pathways such as β-catenin and PI3K/AKT/mTOR, promote cell cycle arrest and apoptosis, and downregulate oxidative phosphorylation [231,232].

Recently, an in vitro study on two OSCC cell lines (SCCL-MT1 and UPSI-SCC-131) reported that Empafliglozin (Emp), an SGLT2 inhibitor, effectively suppresses SLC2A3 and NLRP3 expression, resulting in a notable reduction in tumor cell proliferation and migration [233]. By disrupting glucose metabolism and alleviating chronic inflammation through NLRP3 inhibition, Emp has the potential to lessen the aggressive behavior of OSCC. These insights not only highlight Emp’s promise as a therapeutic agent for OSCC but also underscore the necessity for further research to confirm its clinical efficacy [233].

On the other hand, research indicates that SGLT1 (not SGLT2) is expressed in oral squamous cell carcinoma, correlates with tumor differentiation and a poor prognosis, and may be theoretically relevant for OC [234,235,236]. Still, no clinical trials have been conducted.

### 4.3. Molecular Biomarkers as Therapeutic Targets in Diabetes and Oral Cancer

The intersection of diabetes and OC represents a critical frontier in precision medicine, where shared molecular pathways offer numerous other therapeutic opportunities (Table 6).

## 5. Methodological Limitations and Further Research

While the epidemiological literature consistently demonstrates associations between DM and oral cancer, significant methodological limitations limit causal inference. There is heterogeneity across studies concerning diabetes classification (self-report versus biochemical confirmation), disease duration, treatment regimens, and glycemic control, variables that can influence both metabolic status and cancer risk.

The majority of mechanistic data linking diabetes to oral carcinogenesis derives from in vitro experiments using cancer cell lines exposed to supraphysiological glucose concentrations or from xenograft models that may not recapitulate the chronic, heterogeneous metabolic milieu of diabetes in humans.

Critical evidence gaps include: the absence of longitudinal studies measuring proposed mediators (insulin, IGF-1, inflammatory cytokines, AGEs) before OC diagnosis; limited comparative analyses of pathway activation in oral tumor tissues from diabetic versus non-diabetic patients; and insufficient integration of genetic susceptibility factors that may independently predispose individuals to both conditions. The temporal relationship remains unclear: do diabetes-related molecular changes precede malignant transformation, or do both conditions share common upstream causes, such as chronic inflammation or mitochondrial dysfunction?

While specific pathogens (*P. gingivalis*, *F. nucleatum*, *C. albicans*) are consistently associated with both conditions, establishing causality versus consequence remains challenging. Diabetic hyperglycemia creates favorable conditions for pathogen proliferation; however, periodontal disease itself may worsen glycemic control, creating a bidirectional relationship.

Considering the limitations mentioned above, we suggest further research in these directions:Prospective longitudinal studies measuring proposed molecular mediators (insulin, IGF-1, inflammatory cytokines, AGEs) before cancer diagnosis to establish temporal causality rather than correlation.Direct validation of pathway dysregulation (PI3K/AKT/mTOR, HIF-1α, NF-κB) in human oral tissues comparing diabetic versus non-diabetic patients while controlling for confounding factors like smoking and alcohol use.Microbiome intervention trials testing whether periodontal treatment or antimicrobial strategies reduce cancer incidence in diabetic populations, combined with metagenomic analyses that identify functional pathways beyond taxonomic associations.Mendelian randomization and genetic studies to strengthen causal inference and identify shared susceptibility loci.Biomarker discovery and validation for early detection, specifically in diabetic populations.Randomized controlled trials evaluating enhanced screening protocols and integrated diabetes-dental care models to determine whether intensified surveillance or glycemic optimization strategies could reduce OC morbidity and mortality in this high-risk population.

## 6. Materials and Methods

A comprehensive search of recent literature was conducted across multiple electronic databases, including PubMed/MEDLINE, Embase, Cochrane Library, and Web of Science, from January 2010 to September 2025. The search strategy employed a combination of terms and keywords related to “oral cancer,” “diabetes mellitus,” “molecular biomarkers,” “HIF-1α,” “metabolic pathways,” “inflammatory markers,” “oral microbiota”, “metformin,” antidiabetic drugs,” “GLP-1 receptor agonists,” “SGLT2 inhibitors,” and “therapeutic targets.” Two independent reviewers screened titles and abstracts, with full-text articles retrieved for potentially eligible studies. Reference lists of included studies and relevant systematic reviews were manually searched to identify additional studies.

Due to the heterogeneity in study designs, populations, interventions, and outcomes, a state-of-the-art synthesis approach was employed rather than quantitative meta-analysis. The synthesis was organized thematically around:Molecular mechanisms: Grouped by biomarker categories.Therapeutic interventions: Organized by drug class with emphasis on mechanism of action and clinical evidence.Clinical implications: Integration of molecular insights with therapeutic potential.

For studies reporting quantitative outcomes, effect sizes were extracted and presented with 95% confidence intervals where available.

## 7. Conclusions and Clinical Implications

The present review highlights that the analyzed literature data overwhelmingly support DM as a significant risk factor for OC development. Multiple biological pathways, including hyperinsulinemia, hyperglycemia, chronic inflammation, and immune dysfunction, contribute to this association through complex molecular mechanisms. This relationship is powerful in specific populations and is associated with worse clinical outcomes. Significant clinical implications may involve (a) risk recognition (diabetic patients should be recognized as a high-risk population for oral cancer), (b) enhanced screening (more frequent and thorough OC screening is warranted in diabetic patients), (c) optimal diabetes management (reasonable glycemic control may help reduce OC risk), (d) metformin consideration (metformin therapy may provide additional protective benefits beyond glucose control) and (e) multidisciplinary care (coordination between diabetes specialists and oral health professionals is essential).

The growing evidence establishing diabetes as a risk factor for oral cancer has important implications for clinical practice, public health policy, and future research directions. Healthcare providers caring for patients with diabetes should be aware of this association and implement appropriate screening and prevention strategies, as follows: (1) maintaining an optimal body weight, (2) avoiding destructive habits (alcohol consumption and smoking), (3) regular dental check-ups for diabetic patients, (4) early intervention strategies, and (5) a multidisciplinary approach to managing diabetes and oral health.

## Figures and Tables

**Figure 1 cimb-47-00929-f001:**
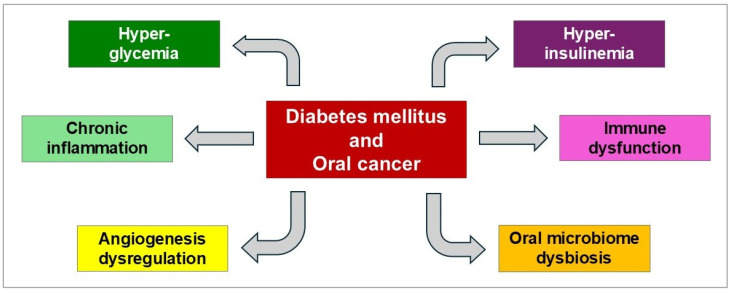
Biological pathways shared by DM and OC [27].

**Table 1 cimb-47-00929-t001:** The impact of DM on OC.

Population	Key Findings	Significance	Reference
- Multiple cohort studies	- Diabetes associated with worse overall survival (HR 1.69, 95% CI 1.29–2.22) - Stronger association in patients ≥52 years old	- Demonstrated negative impact of DM on oral and oropharyngeal cancer prognosis	[24]
- OC patients	- 54.4% of oral cancer patients had elevated glucose levels - 61.1% had Type 2 DM - 4.7% had Type 1 DM	- Evidenced the high prevalence of DM in OC patients	[25]
- OSCC patients	- Lower 5-year survival rate in diabetic patients - Elevated glucose levels increased cancer cell proliferation - Reduced apoptosis and migration	- Demonstrated molecular mechanisms of diabetes impact on OSCC	[25]
- Head and Neck Cancer patients (123 participants)	- Surprisingly, diabetic patients had better progression-free and overall survival - Metformin use is associated with improved survival	- Suggested potential protective effects of diabetes and metformin	[26]
- Diabetic and oral cancer patients	- Minor allele of rs10496915 associated with tumor size - Genetic variations influenced cancer phenotypes in diabetic patients	- Explored genetic factors in the diabetes and oral cancer relationship	[25]
- Tongue Squamous Cell Carcinoma patients	- Lower recurrence rate in diabetic patients - Improved two-year survival compared to non-diabetic individuals	- Suggested potential protective mechanisms	[25]

**Table 2 cimb-47-00929-t002:** AGE-RAGE Interactions: molecular mechanisms and implications in diabetes and oral cancer.

Molecular Mechanism	DM Implications	OC Implications	Molecular Pathways
Inflammation activation	- Chronic hyperglycemia triggers AGE accumulation	- Promotes tumor cell proliferation and invasion	- NF-κB activation
Oxidative stress induction	- Increases ROS production in mitochondria	- Accelerates ROS generation	- MAPK, JNK pathways
Cell apoptosis regulation	- Pancreatic β-cell damage	- Inhibits apoptosis-related proteins	- PI3K/AKT pathway
Insulin resistance mechanism	- Impairs insulin signal transduction	- Alters cellular metabolism	- ERK1/2 signaling
Angiogenesis activation	- Promotes vascular complications	- Enhances cell motility and invasion	- VEGF pathway
TF Activation	- Induces DM - Complications	- Leads to tumor progression and metastases	- IL-6/JAK2/STAT3 pathway

The data displayed in Table 2 are based on [40,55,56].

**Table 3 cimb-47-00929-t003:** Insulin/IGF-1 System in Diabetes and Oral Cancer.

Aspect	Molecular Players	Molecular Considerations
Ligands	- Insulin and IGF-1	- Insulin is produced by pancreatic beta cells in response to blood glucose. - IGF-1 is primarily produced in the liver and is stimulated by GH.
Receptors	- Insulin receptors (IR) - IGF-1 receptors (IGF-1R)	- IR: Primarily binds insulin, linked to metabolic regulation. - IGF-1R: Preferentially binds IGF-1
OC progression	- Stimulation of AKT, PI3K, and Ras-MAPK pathways - Activation of the mTOR signaling complex	- OC cell proliferation, survival, migration, and invasion

All data displayed in Table 3 are based on [63,67].

**Table 5 cimb-47-00929-t005:** Molecular biomarkers with clinical applications in OC and DM.

Biomarkers for Integrated DM–OC Risk Assessment					
Biomarker Type	Marker	DM Association	OC Association	Clinical Significance	Reference
Metabolic	HbA1c	Glycemic control	Increased risk >7%	Risk stratification	[169,170]
	Fasting insulin	Insulin resistance	Mitogenic signaling	Early detection	[171,172]
	IGF/IGFBP3	Growth hormone axis	Proliferation marker	Prognostic value	[173]
Inflammatory	CRP	Chronic inflammation	Tumor promotion	Risk assessment	[174,175]
	IL-6	Cytokine elevation	Oncogenic signaling	Diagnosis and treatment monitoring biomarker	[176]
	TNF-α	Insulin resistance	Tumor stage predictor	Diagnosis biomarker	[177,178,179]
Oxidative	8-OHdG	DNA oxidation	DNA damage	Early diagnosis marker	[180]
	MDA	Lipid peroxidation	Mitochondrial membrane damage	Early diagnosis marker	[181,182]
Genetic	ADIPOQ variants	Adiponectin levels	Cancer susceptibility	Personalized risk indicator	[183]
	TCF7L2 variants	T2DM risk	Cancer metabolism	Genetic marker	[184]
**MicroRNAs Clinical Significance in Diabetes and Oral Cancer**					
**miRNA** **Type**	**Target Genes**	**DM Implication**	**OC Implication**	**Clinical** **Significance**	**Reference**
miR-375	PDK1, YWHAZ	↓ Insulin secretion	↑ Oncogenic in OSCC	Early diagnosis marker	[185]
miR-146a	IRAK1, TRAF6, EGFR	↓ Anti-inflammatory	↓ Tumor suppressor	Prognosis marker	[186,187]
miR-21	PTEN, PDCD4, TPM1	↑ Pro-inflammatory	↑ Oncogenic	Biomarker for diagnosis and prognosis	[186,187,188,189]
miR-126	VEGF, PI3K	↓ Endothelial function	↓ Anti-angiogenic	Vascular complications detector	[190,191]
miR-200 family	ZEB1/2, E-cadherin	↑ Epithelial integrity	↓ EMT suppression	Metastasis marker	[192,193]
let-7	RAS, c-Myc, HMGA2	↑ Glucose homeostasis	↓ Tumor suppressor	Prognosis indicator	[194,195,196,197,198]
miR-34a	CDK4/6, c-Met, Notch	↑ β-cell apoptosis	↓ p53 target	Prognosis indicator	[199,200]

↑ Upregulation; ↓ Downregulation.

**Table 6 cimb-47-00929-t006:** Molecular biomarkers as therapeutic targets in DM and OC.

Biomarker Type	Key Players	DM Interaction	OC Interaction	Therapeutic Targets	Reference
Inflammatory	NF-κB	Chronic activation via AGE-RAGE	Promotes survival, angiogenesis	NF-κB inhibitors, Curcumin	[237,238,239,240]
	TNF-α	Insulin resistance	Tumor promotion, cachexia	Anti-TNF antibodies	[241,242]
	IL-6	β-cell dysfunction	Oncogenic signaling	Tocilizumab	[243,244]
ROS-related	ROS/RNS	Pancreatic damage	DNA damage, mutagenesis	Antioxidants, Nrf2 activators	[245,246]
	NADPH Oxidase	Vascular complications	Pro-tumorigenic signaling	NOX inhibitors	[247,248,249]
	SOD/Catalase	Reduced activity	Impaired DNA repair	Antioxidant enzymes	[250,251]
Metabolic	mTOR	Nutrient-sensing disruption	Growth promotion	Rapamycin, mTOR inhibitors	[252,253]
	AMPK	Reduced activation	Tumor suppression loss	AMPK activators	[254]
	Insulin/IGF-1	Hyperinsulinemia	Mitogenic signaling	IGF-1R inhibitors	[255,256]
Angiogenic	VEGF	Diabetic complications	Tumor angiogenesis	Bevacizumab, VEGF inhibitors	[257,258]
	HIF-1α	Hypoxia response	Metabolic reprogramming	HIF inhibitors	[259,260,261]
Immune	Th1/Th2 balance	Shifted to Th2	Tumor immune evasion	Immunomodulators	[262,263,264,265]
	PD-1/PD-L1	Immune exhaustion	Checkpoint inhibition	Pembrolizumab, Nivolumab	[266,267]

## Data Availability

No new data were created or analyzed in this study.

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
