# Peer review of "Interconnected Mechanistic Pathways, Molecular Biomarkers, and Therapeutic Approach of Oral Cancer in Patients with Diabetes Mellitus"

_cimb, 2025, doi:10.3390/cimb47110929_

Round 1
Reviewer 1 Report
Comments and Suggestions for Authors 1. In section 2, "Biological mechanisms that correlate diabetes mellitus with oral cancer," a generic list of related mechanisms is presented. These mechanisms are mentioned but not elaborated upon, which is partially redundant with section 3, "Molecular Pathways in Diabetes and Oral Cancer."2. A grammatical review is necessary; the manuscript presents multiple redundancies and a lack of depth of ideas throughout the text, as well as formatting errors, for example, the use of unnecessary capital letters (line 134, "Elevated levels"). Another example of a lack of adequate depth is the subtopic "3.2. Core Molecular Pathways," Table 3, which is a wealth of information. However, placing it before the descriptive text makes it chaotic.
3. Consider whether it is necessary to include Figure 1; it does not add more information to what has already been written. It lacks proper correspondence, as it primarily refers to the content of Section 2 (except for the terms' immune dysfunction' and 'metabolic reprogramming'). If you decide to keep it, the design should be improved. Placing the terms' diabetes mellitus' and 'oral cancer' in the central box of shared molecular pathways would give a simpler appearance.
4. In Table 3, review the involvement of the inflammatory pathway; for example, for NLRP3 and ASC, the table highlights one function, while the descriptive text highlights others. The same applies to the p53 Tumor Suppressor Network.
5. A better sequence is needed in the wording of the GLP-1 agonist to justify its inclusion in the review. It should describe that these are new therapeutic strategies for the treatment of diabetes; however, this information is very vague. Comments on the Quality of English Language
The manuscript requires a grammatical revision
Author Response
Dear Reviewer 1,
We are grateful for the accurate peer review process, which substantially improved the quality of the revised manuscript. We thank Reviewer 1 for their time, professionalism, and valuable comments. We responded point by point to each comment and marked the changes in red. Moreover, we reformulated the paragraphs with a high similarity index and marked them in green.
Please, see the response in the attachment.
Thank you again for everything.

Reviewer 2 Report
Comments and Suggestions for Authors
Elian et al studied the literature and wrote the review on incidence of oral cancer (OC) in Diabetes mellitus (DM) patients. DM patients have numerous immune mediated complications that induces OC occurrence which is increased in DM. They also emphasized that metformin reduces the OC by modulating mTOR pathway and GLP1 agonist may also improve the condition, although further investigation is necessary.
This paper is a timely documentation about the area, but some concerns need to be fixed. Authors should edit the manuscript thoroughly.
- In line 35 and line 64, prevalence (0.25%) does need more explanation.
- Line 260, P. gingivalis, P is abbreviated but again in line 267, it is expanded. Authors should follow consistency.
- Line 260, 275, same is for Fusobacterium.
- Line 351-352, sentence is incomplete.
- Line 367-374, in the text, sometimes diabetes and sometime T2DM or DM, authors should have consistency.
- Sometimes oral cancer is mentioned as abbreviated (OC)and sometimes OSCC! However, OSCC is not in the list.
Author Response
Dear Reviewer 2,
We are grateful for the accurate peer review process, which substantially improved the quality of the revised manuscript. We thank Reviewer 2 for their time, professionalism, and valuable comments. We responded point by point to each comment and marked the changes in red. Moreover, we reformulated the paragraphs with a high similarity index and marked them in green.
We've attached the response.
Thank you again for everything.

Reviewer 3 Report
Comments and Suggestions for Authors
This manuscript offers a comprehensive and well-organized overview of the molecular and clinical relationship between diabetes mellitus (DM) and oral cancer (OC). The authors integrate complex biochemical, genetic, immunological, and therapeutic data.
The authors bring something new: the bidirectional nature of DM–OC interaction.
Suggestions:
- Figure 1 needs improvement, like arrows for a better understanding.
- Make a table for clinical and epidemiological studies with headings like type of study, population, findings, etc.
- Minor English revision.
- Refine conclusions: The Essential Considerations section could be retitled Conclusions and Clinical Implications. Strengthen a practical message with recommendations.
Minor English revision.
Author Response
Dear Reviewer 3,
We are grateful for the accurate peer review process, which substantially improved the quality of the revised manuscript. We thank Reviewer 3 for their time, professionalism, and valuable comments. We responded point by point to each comment and marked the changes in red. Moreover, we reformulated the paragraphs with a high similarity index and marked them in green.
We've attached the response.
Thank you again for everything.

Round 2
Reviewer 1 Report
Comments and Suggestions for Authors
The manuscript presents the necessary and sufficient corrections that improve their presentation.